# The Role of Iron in DNA and Genomic Instability in Cancer, a Target for Iron Chelators That Can Induce ROS

Andrew Carter *, Seth Racey * and Stephany Veuger *

Department of Applied Sciences, Faculty of Health and Life Sciences, Northumbria University, Newcastle upon Tyne NE1 8ST, UK
* Correspondence: andrew.w.carter@northumbria.ac.uk (A.C.); seth.racey@northumbria.ac.uk (S.R.); s.veuger@northumbria.ac.uk (S.V.)

**Abstract:** Iron is a key metal involved in several biological processes such as DNA replication and repair, cellular proliferation and cell cycle regulation. Excess volumes of labile iron are toxic and can lead to the production of ROS (reactive oxygen species) via Fenton chemistry. Due to this reactive nature, it can contribute to DNA damage and genomic instability. Therefore, excess iron in the labile iron pool is associated with cancer, which has made the labile iron pool a crucial target for anticancer therapy by targeting iron. This iron can be incorporated into essential enzymes such as ribonucleotide reductase (RnR). Over several decades of research, iron chelators function as more than just RnR inhibitors. Indeed, a plethora of iron chelator mechanisms can result in therapeutic properties that can target critical steps of cancer cells' aberrant biological abilities such as proliferation, migration and metastasis. One such mechanism is the production of redox-active complexes that can produce toxic levels of ROS in cancer cells. Cancer cells are potentially more susceptible to ROS production or modulation of antioxidant levels. Understanding iron metabolism is vital in targeting cancer. For instance, Fe-S clusters have recently been shown to play crucial roles in cell signalling by ROS through their incorporation into essential DNA replication and repair enzymes. ROS can also degrade Fe-S clusters. Iron chelators that produce toxic levels of ROS, therefore, could also target Fe-S centres. Thus, the design of iron chelators is important, as this can determine if it will participate in redox cycling and produce ROS or if it is solely used to remove iron. This review focuses on alterations in cancer iron metabolism, iron's role in genomic stability and how the design of chelators can use Fenton chemistry to their advantage to cause DNA damage in cancer cells and potentially inhibit Fe-S centres.

**Keywords:** iron; iron chelation; cancer; drug design; Fe-S centres; DNA damage; ROS

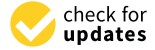



## 1. Introduction

Iron is the most abundant transition metal in the human body and a vital micronutrient that is a critical component of many crucial enzymes. Therefore, it is essential to various biological processes, such as DNA synthesis and repair, cell cycle regulation, transport of oxygen, and energy production [1]. Iron's catalytic ability harnessed in these crucial enzymes lies in iron's potential to change between various oxidation states, the di-, tri- and, less frequently, the tetravalent oxidation states. The potent redox potential of iron allows it to cycle between these di-, tri- and tetravalent states, which makes it crucial in electron transfer and oxygen transport [2]. However, this strong redox potential allows Fenton chemistry (Figure 1), leading to the production of free radicals and reactive oxygen species (ROS) such as the hydroxyl-radical HO$^\bullet$, the superoxide radical O$_2^{\bullet-}$, and hydrogen peroxide H$_2$O$_2$. The free radicals created by the Fenton reaction results in subsequent damage to macromolecules such as DNA, lipids and proteins [3]. When iron is in excess, Fenton chemistry is enhanced, producing ROS, generating organic radicals and aldehydic by-products of lipid and amino acid oxidation. The overproduction of ROS can result

in DNA and mitochondrial damage and genetic instability in the cell. Therefore, iron chemistry must be tightly regulated in normal cells to prevent cytotoxicity and normalise cell functions and growth.

$$Fe^{3+} + O_2^{\cdot-} \rightarrow Fe^{2+} + O_2 \tag{A}$$

$$Fe^{2+} + H_2O_2 \rightarrow Fe^{3+} + OH^{\cdot} + OH^- \tag{B}$$

$$O_2^- + H_2O_2 \rightarrow O_2 + OH^{\cdot} + OH^- \text{ in the presence of a Fe catalyst} \tag{C}$$

**Figure 1.** The reduction of Fe (III) by the superoxide radical (**A**), the Fenton reaction (**B**) and the complete oxidation-reduction process are known as the Haber–Weiss reaction (**C**).

Keeping iron levels safe and constant is crucial as conditions of excess or deficiency can lead to diseases such as iron-deficient anaemia and Haemochromatosis, respectively. However, an iron imbalance intracellularly can also lead to neoplastic disease as iron is vital for DNA synthesis and repair, both of which are needed to prevent cancerous changes. For instance, a lack of intracellular iron regulation can result in changes that enhance proliferation, one of the key hallmarks of cancer [4,5]. A wide variety of iron metabolism proteins play an essential part in tumour progression, the functions of which are required by cancerous cells at relatively higher levels than that of normal cells. Intracellularly iron is commonly bound in heme groups, Fe-S clusters and ferritin. Iron in the form of $Fe^{2+}$ is a heme cofactor in both myoglobin and haemoglobin, which are important in oxygen transport in the body. whereas iron clusters participate in electron transfer, play a structural role in DNA enzymes and play iron regulatory roles such as controlling gene expression in response to iron [6]. Iron clusters, [2Fe-2S], [4Fe-4S] or [3Fe-4S], frequently contain an iron centre of $Fe^{2+}$ or $Fe^{3+}$. These iron clusters play important structural and enzymatic roles in DNA replication and repair and are key components of DNA helicases, primases and polymerases. While iron can be found in these iron clusters, some iron is also present in the cell as a weekly chelate-able pool called the labile iron pool, also known as the redox-active pool or free iron pool. The labile iron pool is mainly a $Fe^{2+}$ pool with ions bound to low-affinity ligands; it is this free $Fe^{2+}$ that is used in intracellular iron metabolisms and reactions. Thus, the labile iron pool is important in Fenton chemistry and the consequent production of harmful radicals, particularly when the labile iron pool is in excess. Many studies show an increased labile iron pool in various cancerous cells and an increased ferritin expression; however, a direct cause-and-effect relationship has yet to be fully elucidated [7,8].

Hence, iron chelators are an attractive therapeutic option for restoring normal iron homeostasis and consequently inhibiting cell growth and intracellular ROS formation by removing iron from the labile iron pool and DNA enzymes [9]. Most cancer cells have a much higher basal level of ROS due to the increased labile iron and an increased level of antioxidants, which often counterbalance this increase in ROS [10]. This counterbalance can thus result in constantly increased levels of ROS not being toxic to the cells. However, cancerous cells can be killed by compounds that increase ROS; these can tip the balance, resulting in lethal levels of DNA damage. Iron chelators can be designed whereby their complexes participate in redox cycling, thus increasing ROS levels to a toxic level in cancer cells. These iron chelators can cause DNA damage and kill cancer cells and have recently been used in combination therapy for cancer [11,12]. This review will seek to elucidate iron's role in metabolism and how this is altered in cancer. Therefore, the review will touch on iron chelators that can limit the labile iron available for incorporation into vital Fe-S clusters used in DNA replication and repair. Lastly, the review will focus on the vital role of iron in DNA replication and repair pathways and the design considerations that need to be considered so that iron chelators can be designed to take advantage of cancer cells' susceptibility to toxic levels of ROS. This review will not cover iron chelators and

cancer stem cells; we refer the reader to additional reviews for the complete picture of iron chelators, metabolism and cancer stem cells [13,14].

## 2. Iron Chelators Design and Redox Activity of Iron

### 2.1. Redox Activity

ROS is produced during aerobic respiration and involves free iron being utilised in the Fenton and Haber Weiss reactions, as shown in Figure 1 below.

ROS levels are typically under tight regulation and are used beneficially in cell signalling using iron-dependent enzymes for cell survival and proliferation [15,16]. However, if the reactions are uncontrolled, they can produce toxic levels of ROS, which can result in oxidative damage and may contribute to carcinogenesis [17,18]. Thus, iron's pathological importance is due to this redox activity enabled by its ability to participate in Fenton chemistry, which results in ROS production. Redox cycling can occur if the oxidised complex is reduced, and the reduced complex can give an electron. The design of a chelator is important as chelators that can form redox-active metal complexes can produce cytotoxic ROS, killing cancerous cells. Another critical design consideration is thermodynamics, which predicts the direction of free radical reactions and the complexation of a metal centre, such as in the Fenton reaction and Haber Weiss reactions. The electromotive force is vital when calculating free radical reactions. In particular, the half-cell reduction potential shows if the reaction will occur in standard conditions, where the cell is referred to as a galvanic cell. The electromotive force for two half cells and Gibs free energy shows that the Fenton reaction is favourable in Figure 2. The term Gibbs free energy has been changed, favouring the term electrode potential defined by the IUPAC [19].

$$\Delta E = E2 \text{ (acceptor of electrons)} - E1\text{(donor of electrons)} \qquad (A)$$

$$\Delta G° = \Delta H° - T\Delta S° = RTlnK = -nF\Delta E \qquad (B)$$

**Figure 2.** Part (**A**) The electromotive force where E2 is the reduction potential for the half-cell reaction of the reduced species. E1 is the reduction potential for the half-cell reaction of the species oxidised if the cell's potential is negative. The reaction can be reversed. Part (**B**) shows Gibbs free energy of the reaction (kJ mol$^{-1}$), $\Delta H°$ is the standard enthalpy of the reaction (J mol$^{-1}$), T is the temperature (K), and $\Delta S°$ is the standard entropy of the reaction (JK$^{-1}$ mol$^{-1}$), R is the gas constant (J/mol K), K is the equilibrium constant of the reaction, F is the Faradays constant = 96,494 (C/mol), and n is the number of electrons involved in the transfer.

Using the above equations, we see that the electrode potential of the hydrogen peroxide/hydroxyl radical (Eo = +0.32V, pH 7, 25 °C, NHE) is favourable with respect to Fe$^{3+}$/Fe$^{2+}$ (Eo = +0.77 V, pH 7, 25 °C, NHE). Thus, it is likely to oxidise, producing ROS in standard conditions [20,21]. However, the electrode potential of Fe$^{3+}$ (complex) + e can vary significantly depending on the type and strength of ligand from −1 V to +1 V [22]. Consequently, the ligands are essential in the coordination sphere of Fe$^{2+}$, as the reactivity depends on the Fe complexes' speciation an important consideration in iron chelator design [23]. Iron chelators that can bind to both Fe (II) and Fe (III) can form redox-active complexes; this facilitates redox cycling resulting in the production of more toxic free radicals [24]. Iron chelators, such as traipine that form redox-active complexes, can produce toxic ROS levels, killing the cancer cells [25,26]. Therefore, the design of a chelator is important in terms of its redox activity. The design determines if it can act in one of two ways: first, the increased production of ROS by using a redox-active chelator, or secondly, no production of ROS with the chelator's sole purpose being to remove the desired metal. Thus, the second design consideration is to act as a non-redox-active chelator. The second design consideration is of much more use in iron overload diseases where the production of ROS would be undesirable. Iron chelators may also affect physiochemical properties, not just directly interacting with redox reactions however resulting in a change to the electrode potential of the iron complex. The resulting iron complex may have free binding sites that

other ligands can bind to or that ROS can interact with, generating further ROS. The redox activity of the resulting complex is multilayered and should have great consideration when designing iron chelators for a specific purpose.

### 2.2. Thermodynamics, Stability, Sensitivity and Lipophilicity

The selectivity and stability of the chelator complex are essential when designing a chelator. The ligands, as discussed above, can change the electrode potential of iron complexes but can also influence the stability of the complex. Chelators must form at least two (bidentate) coordination bonds to the metal, and iron can achieve up to a maximum of six (hexadentate) coordination bonds. They usually contain ligands of nitrogen, oxygen and sulphur [26]. The number of chelate rings affects stability, with an increasing number resulting in increased stability; this is the chelate effect [27]. Thermodynamically, this is favourable, as it has entropic and enthalpic contributions due to the increase in the strength of the binding site caused by an increase in electron density. Thus, the binding of a chelator with a hexadentate coordination number is also redox stable as it saturates the Fe coordination sphere and binds iron in a (1 chelator: 1 iron) ratio. The redox stability of hexadentate chelators can be proven as hydroxyl radical formation requires a free unbonded coordination site in Fenton chemistry; therefore, hexadentate chelators do not perform Fenton chemistry [24,28]. Iron chelators such as bidentate chelators bind iron in a (3 chelator: 1 iron) ratio and tridentate chelators in a (2 chelator: 1 iron) ratio, which can leave co-ordination sites free [29]. The bidentate and tridentate chelators are more kinetically labile as they can form these partially dissociated complexes. For example, a bidentate chelator might only bind to iron twice, leaving a site free. This results in the weaker chelators that might not bind with all the coordination sites being more reactive, increasing Fenton chemistry and ROS [26]. Bidentate, tridentate and hexadentate chelators can be seen in Figure 3, all occupying a full coordination site. Therefore, the stability of these ligands goes in the order of hexadentate > tridentate > bidentate [30]. The stability of these compounds can be ascertained by looking at the $pFe^{3+}$ values, which is the negative logarithm concentration of free Fe (III). This also considers the ligand protonation, denticity, and stoichiometries and shows that hexadentate chelators are more stable than bidentate and tridentate ligands [31].

Thermodynamic stability depends on the bonding interaction, where the metal-ligand interaction can be described as Lewis acid–base interaction [32]. Chelators can be designed for either Fe (III) or Fe (II). High spin Fe (III) acts as a hard acid as it has a tri-positive cation ratio of 0.65 Å, a high electron density, and binds hard ligands such as oxygen. Oxygen donor atoms can also stabilise the chelator, and thus it is much less likely to redox cycle [33]. In contrast, Fe (II) has a lower electron density and prefers softer ligands such as nitrogen, which can be reduced in physiological conditions by enzymes in the cell [30]. Due to this redox activity, they can participate in Fenton chemistry and generate ROS. It is important to note that ligands that prefer Fe (II) have an affinity for other bivalent metals [34]. Contrastingly ligands that prefer Fe (III) cations prefer cations such as gallium that are not commonly seen in the human body. This is why Fe (III) chelators are becoming more popular in a clinical setting for iron overload diseases [35]. Ligands that bind both are more likely to favour Fe (II), which prefers soft donors. These soft donors would be more likely to catalyse the Fenton reaction leading to the production of hydroxy radicals and are an attractive option for cancer by increasing ROS [29].

Lipophilicity refers to the affinity a molecule has for an aqueous environment, usually determined by its log *p*-value, the logarithm of solute concentration in octanol over union-ized solute concentration in water [36]. Lipophilicity is crucial as it allows the uptake of the drug in the gastrointestinal tract to reach the target site to release its therapeutic effect [37]. Lipophilicity is essential when designing a chelator, as it can utilise passive transport to cross biological barriers. There are two routes an oral drug such as an iron chelator can take either through the portal vein or the intestinal tract. If they are small and soluble, they can travel through the portal vein, where enzymes metabolise them in the liver [38]. The

second is through the lymphatic system, bypassing the metabolism in the liver, increasing its bioavailability [39]. Lipophilicity is vital for it to pass through the lymphatic system. The addition of lipids, or a more lipid ligand, can help an iron chelator's bioavailability in three ways: the alteration of the intestinal milieu, recruitment of the lymphatic drug transport system and interaction with the enterocytes transport process [2,40]. In theory, chelators with high lipophilicity can enter the cells more efficiently and therefore bind to the labile iron inhibiting it from many processes such as direct inhibition of the R2 subunit of ribonucleotide reductase (RR) [41]. As discussed above, the size of the chelator is also important since if it is above a specific molecular weight, roughly 500 Daltons, it will not be absorbed through the gastrointestinal tract. Additionally, most hexadentate chelators have more than 5 high hydrogen bond donors and 10 hydrogen bond acceptors, resulting in poorer absorption or permeation [42]. This limits which chelators can be used; typically, hexadentate chelators such as bidentate and tridentate have lower molecular weights. Therefore, they are more orally available than hexadentate chelators which struggle to be absorbed and enter cells. An example of this is deferoxamine; a hexadentate chelator that does not quickly enter or leave cells [43].

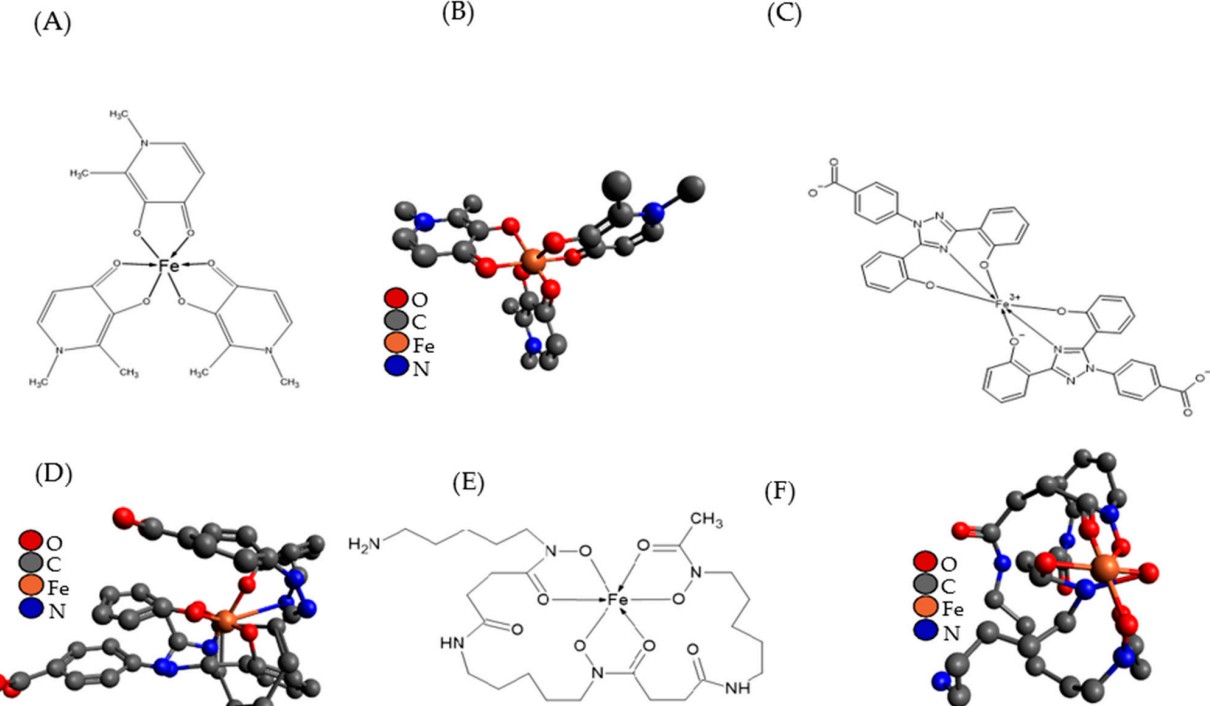

**Figure 3.** (**A,B**) show deferiprone, the bidentate chelator, bound to iron occupying its full coordination in a 3:1 ratio (3 deferiprone: 1 iron) and its 3D arrangement are, respectively, optimised. (**C,D**) show the tridentate chelator deferasirox bound to iron in a ratio of 2:1 (2 deferasirox to 1 iron). These chelators are more likely to be redox-active than hexadentate chelators as they can leave sites free for Fenton chemistry. (**E,F**) show the hexadentate chelator bound with iron Ferrioxamine B. Ferrioxamine is not redox-active, as it binds iron 1:1 and therefore cannot leave any sites free.

Therefore, the design of a chelator for its use is clinically significant. Chelators that should be used in iron overload diseases should be redox inactive and are therefore typically hexadentate ones. However, the production of ROS species by iron chelators, where their complexes participate in redox cycling, has been shown to kill cancer cells by causing DNA damage. Consequently, bidentate or tridentate iron chelators with soft donors can typically bind to both $Fe^{2+}$ and $Fe^{3+}$. They are an attractive option as they can form complexes that redox cycle and produce excess ROS which can be toxic to cancer cells. Bidentate and

tridentate chelators are less stable; however, this ability to develop dissociated complexes with iron allows its kinetic lability and participation in Fenton chemistry.

### 3. Brief Overview of Iron Metabolism, the Labile Iron Pool and How It Is Altered in Cancer

#### 3.1. Cellular Iron Homeostasis

At physiological pH and oxygen concentration, iron(II) in the body is readily oxidised to iron(III); as a consequence of this, iron is chaperoned to prevent it from partaking in undesirable chemical reactions such as the Haber–Wiess reaction [44]. Once iron has been taken in by the digestive tract, iron is then transported in the plasma by transferrin (Tf). Transferrin binds two ferric ions ($Fe^{3+}$) and has a high affinity for iron. The iron–transferrin complex binds to transferrin receptor 1 (TfR1) on the surface of a cell that expresses it. The di-ferric transferrin enters the cell by receptor-mediated endocytosis. Once released, iron is reduced by ferrireductase before transportation across the endosomal membrane by the divalent metal transporter1 (DMT1) [45]. Once iron is in the cytoplasm, it is stored in a complex with ferritin (Ft) or used to form Fe-S clusters or heme proteins or contribute to the labile iron pool [46]. Iron can also be transported out of the cell by ferroportin (FPN1). Iron which is neither stored nor transported out of the cell exists in several organelles as the labile iron pool.

The labile iron pool is in a dynamic equilibrium, shifting and changing its constituents depending on the availability of iron, resulting from the influx and efflux of iron from the cell or the stored forms of iron. The iron within this labile iron pool can either be $Fe^{2+}$ or $Fe^{3+}$ but is generally accepted to contain more $Fe^{2+}$ due to the great quantity of water and reductants in cells [47]. Iron within the labile iron pool is bound to intracellular proteins to minimise potentially toxic Fenton chemistry. One of the main intracellular proteins is iron(II)glutathione conjugates [47,48]. The labile iron pool is generally regulated by storing excess iron in a complex with ferritin or exporting it outside the cell with FPN-1. One of the main ways of protecting normal cells from excess labile iron and Fenton chemistry is storing iron in ferritin. For exporting, $Fe^{2+}$ iron is converted back to $Fe^{3+}$ by ferroxidase [49]. The molecules Ft and FPN-1 have an iron response element: hairpin loop structures in the 5′ direction on their untranslated region (UTR) of mRNA [50]. These interact with mRNA binding molecules called iron-regulating protein one and iron-regulating protein two (IRP1 and IRP2). When iron conditions are low, IRP1 binds to an iron-responsive element (IRE) in the 5′UTR on the mRNAs of Ft or FPN-1, which blocks, through steric hindrance, the recruitment of ribosomes and obstructs translation [51]. IRP1 binds to the 3′ UTR on TfR1, and DMT1 masks the mRNA to endonuclease digestion, stabilising it. The consequence of the IRP1s actions on iron is to reduce export and increase import, increasing iron levels in the cell. This occurs as IRP1 is an enzyme known as cytosolic aconitase and contains a full [4Fe-4S] cluster in the presence of high iron conditions. However, when intracellular iron levels are low, there is not enough iron for Fe-S biogenesis resulting in a [3Fe-4S] cluster on the enzyme [52]. Therefore, aconitase cannot function enzymatically, resulting in its IRP1 activation. IRP1 therefore utilises a unique iron-sulphur cluster switch to sense iron levels. Simultaneously, low iron conditions result in the ubiquitination of IRP2, resulting in more iron efflux [50]. Through these mechanisms, in normal conditions, the labile iron pool size results from increases or decreases in iron influx and efflux, which are regulated in correspondence to the cells' needs.

Further to IRP1, iron homeostasis can also be maintained by hepcidin through several mechanisms: it is a hormone that induces FPN-1 degradation, can inhibit recycled iron release from macrophages, absorption in the duodenum and lastly, through iron stored in hepatocytes [53]. Hepcidin expression correlates with both cellular and serum iron levels. When iron levels are high, hepcidin is produced in the liver. This consequently results in the degradation of FPN-1 in duodenal enterocytes and hepatocytes, which prevents these cells from exporting iron [50]. In contrast, this process does not occur when iron levels are low, and the duodenal and hepatocytes can export iron [54].

Mitochondrial Iron Metabolism and Fe-S Biosynthesis

Iron plays a critical role in the mitochondria; however, the exact mechanisms of how the mitochondria receive iron have not been fully elucidated. Mitochondrial iron is the leading destination for cytosolic labile iron. Thus, cytosolic iron from the labile iron pool must cross the outer mitochondrial membrane (OMM) before being utilised for Fe-S cluster synthesis within the mitochondria with its own labile iron pool. The mitochondrial labile iron pool is also susceptible to ROS as the mitochondria are a site of oxygen consumption. There are two hypotheses as to how mitochondrial iron crosses the OMM. The first is that STEAP3 reduces $Fe^{3+}$ iron in the endosome [55]. The second is the kiss and run theory, where there is the docking of the endosome containing transferrin-bound iron [56]. More evidence is needed to clarify the exact mechanism or mechanisms involved for iron to cross the OMM. However, the mechanisms for iron to travel from the OMM to the inner mitochondrial membrane (IMM) have been elucidated. It involves iron being brought across the IMM by mitoferrin 1 and mitoferrin 2. It has been shown that purified recombinant mitoferrin transports free iron and that its reduced expression results in iron depletion [57,58]. Once transferred across the IMM by mitoferrin 1 and mitoferrin 2, it is incorporated into Fe-S clusters by frataxin and GLRX5 (Glutaredoxin-related protein 5) for many processes, including DNA repair enzymes [59].

The mitochondrial iron sulphur cluster (ISC) assembly machinery starts Fe-S biogenesis. This complex multistep process can be broken down into four main steps. It is crucial not only for the maturation of Fe-S clusters in the mitochondria but also for cytosolic and nuclear Fe-S cluster proteins [60]. The first [2Fe-2S] cluster is synthesised de novo by several vital ISC proteins; this takes place on the ISU1 scaffold protein [61]. Next, an ATP-dependent chaperone protein aids the Fe/S cluster release from Isu1 and its transfer to monothiol glutaredoxin [62]. It can then be used in proteins that require a [2Fe-2S] cluster or passed to the ISC for [4Fe-4S] cluster synthesis; the latter is the third step. Lastly, the fourth step involves the insertion of the [4Fe-4S] cluster into apoproteins that require it by ISC machinery [63]. The labile iron pool has crosstalk between the mitochondrial labile iron pool and the cytosolic, where the demand for Fe-S clusters and heme synthesis in the mitochondria can lower the cytosolic levels of labile iron. The labile iron in both the cytosol and mitochondria that iron chelators target theoretically makes less iron available for incorporation into crucial iron-requiring enzymes.

### 3.2. Altered Iron Metabolism in Cancer

Cancer cells have a higher requirement for iron in order to have a proliferation advantage, which is a cancer hallmark [4,5]. Thus, cancers have adapted to increase the overall iron content in the cells. They have achieved this adaption by increasing iron intake and decreasing the efflux, for example, by increasing the transferrin receptor and decreasing ferroportin [64–66]. The increase in transferrin can be the result of the oncogene c-Myc. C-Myc is essential as it can be transitionally upregulated in TFR. Alternatively, it can result from hypoxia as iron is needed to deliver oxygen to cells in low oxygen conditions or by IRP2 upregulation [67,68]. Many critical enzymes involved in importing iron are upregulated in cancer, such as membrane receptors such as DMT1, which are overexpressed in several cancers [69]. Proteins that facilitate endosomal uptake of the di ferric transferrin complex are also increased in cancer, and the subsequent STEAP 1–4 proteins that are involved in reducing iron in endosomes are also overexpressed in cancer [46,70–72].

As stated above, cancer cells can limit the volume of iron that can efflux; an example of this is the downregulation of ferroportin in several cancers such as ovary, lung myeloma and many more [73–75]. Hepcidin can also modulate ferroportin by binding to it, resulting in its eventual degradation; this has shown to be upregulated in prostate cancer cells and shows one method in which ferroportin can be decreased in cancer cells [76]. Irons decreased efflux in cancer cells and increased influx results in a much-increased labile iron pool, as shown in Figure 4. This satisfies its need for iron for the essential process, resulting in further genomic instability by iron-induced ROS. The increased labile iron

pools in cancer allow more Fe-S synthesis for crucial cell functions such as DNA repair and replication. Simultaneously, it potentially results in more genetic instability by enhanced Fenton chemistry, aided by the abundance of iron. Therefore, iron chelators have been designed to target labile iron pools, remove iron, and disrupt one of cancer cells' main adaptive mechanisms.

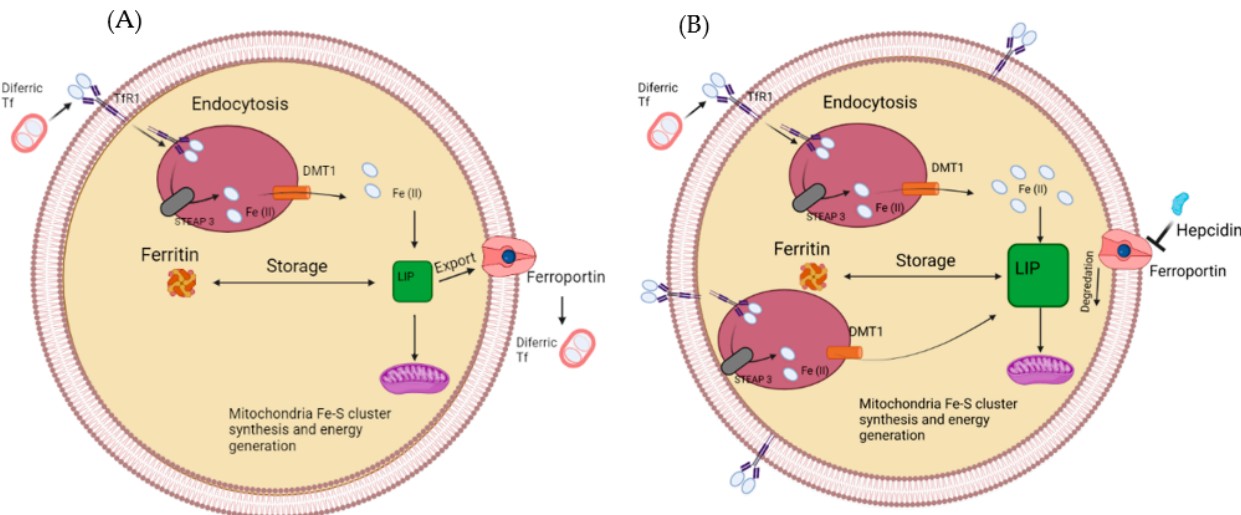

**Figure 4.** (**A**) The labile iron pool in a normal healthy cell. Iron enters via the transferrin receptor before getting reduced and used for storage or the labile iron pool (**A**). (**B**) How the labile iron pool is altered in cancer cells, upregulated TfR1, STEAP3 and DMT1, while a downregulated Ferroportin results in a larger labile iron pool.

## 4. Genomic Instability, Fe-S Clusters and Iron Chelators

The physical structure of the genome consists of DNA molecules wrapped around histone proteins that are then assembled in chromosomes. DNA divides in a carefully monitored process controlled by the DNA damage response (DDR) machinery. To maintain genomic stability there needs to be accurate replication and repair; if this cannot be achieved, it can lead to cancer development. As described earlier, aberrant levels of iron can increase ROS via Fenton chemistry. The increased ROS can cause DNA damage which implicates iron in genomic instability. In addition, iron is also involved in DNA replication and repair as Fe-S clusters are integral parts of enzymes involved in the DDR. Key enzymes involved include DNA polymerases (Pol $\alpha$, Pol $\delta$ and Pol $\varepsilon$), the DNA helicases, primases and glycosylases, which all have Fe-S clusters. Notably, the redox activity of Fe-S clusters in these enzymes has also been proposed to play crucial roles in the biochemical and cellular functions [77]. Fe-S clusters are vulnerable to oxidation due to their redox ability [63]. Fe-S clusters can play a structural or active role in DNA damage repair, and their abundance in DDR suggests that they are vital for their function.

Recent studies have shown possible links between Fe-S metabolism and cancer. One possible link is evidence that the overexpression of microRNA miR-210 is involved in suppressing ISCU in breast cancer solid tumours [78]. ISCU and frataxin are controlled at the transcription level by p53, which controls the volume of ROS in cells. Therefore, changes in Fe-S metabolism could be crucial in cancer progression. Other Fe-S cluster proteins that have shown links to cancer are NEET proteins, such as nutrient-deprivation autophagy factor-1 (NAF-1), which are overexpressed in cancer and are also thought to make the cancer cells resistant to oxidative stress [79]. It is clear that iron is essential in genomic stability through the enzymes involved in the DDR and that aberrant iron levels could also lead to tumorigenesis and cancer. Further, as iron levels are elevated in cancer cells and DNA is required at a higher volume, the ability to remove chelate-able iron in cancer could restrict DNA replication. Even so, as some iron chelators can induce ROS, it could also cause DNA damage to the cancerous cells by inhibiting the volume of iron

available or potentially by inactivating the iron enzymes involved in the DNA damage response. More recently, iron chelators have been shown to inhibit the DNA repair protein ALKBH2 by removing available iron that is critical for catalytic function [80].

*4.1. Iron Is Vital for DNA Replication and Repair*
DNA Helicases and Polymerases

DNA helicases contain Fe-S clusters in the N terminus, essential in preserving DNA stability. DNA helicase enzymes include XPD, FANCJ, Dna2, Pud3, RTE and ChlR1 [81,82]. XPD plays an integral part in nucleotide excision repair as a component of transcription factor II H (TFIIH) [83]. XPD catalyses the transcriptional site of DNA duplex opening at the transcriptional site damage or the site of DNA damage [84]. XPD contains a 4Fe-4S cluster and is essential in genome maintenance through its role in nucleotide excision repair. Mutations in XPD cause trichothiodystrophy; this is due to mutated-XPD being unable to bind to single-stranded DNA and thus repair DNA by nucleotide excision repair [84].

FANCJ is a helicase that functions as a tumour suppressor caretaker and iron-containing protein: it contains a Fe-S cluster in a domain that has a vital role in regulating the enzyme. Additionally, it was indicated as a tumour suppresser caretaker as mutations were prevalent in breast cancer [85]. FANCJ is important in double-strand break repair by its direct interaction with BRCA1 through BRCT motifs; it also plays a role in homologous recombination and the Fanconi anaemia (FA) pathway of interstrand crosslink repair [86]. FANCJ helps to maintain chromatin structure and resolves G-quadruplex structures [87]. Mutations in FANCJ contribute to Fanconi anaemia, a disease where the patients are cancer-prone due to a hypersensitivity to agents that cause DNA crosslinks [88]. Further molecular studies are required to see the precise molecular functions of FANCJ.

ChlR1 is a member of the FANCJ and XPD family and has a crucial 4Fe-4S cluster [89]. Mutations in ChlR1 cause Warsaw Breakage syndrome, which leads to multiple abnormalities, this genetic instability syndrome predisposes people with this condition to cancer [90]. Recently, ChlR1 was shown to participate in DNA repair pathways. In a study where ChlR1/DDXII knockdown Hela cells were used with cisplatin, the Hela cells were highly sensitive to cisplatin [91]. Interestingly, cisplatin inhibits iron-responsive element-binding protein (IRP2), causing iron depletion similar to a chelator [92]. Studies also show ChRl1 role in normal sister chromatid cohesion [93].

Dna2 is an enzyme used in telomere maintenance, DNA double-strand break repair, Okazaki fragment processing, and cell cycle activation [94]. Dna2 also contains a 4Fe-4S cluster essential for replication and repair [95]. Furthermore, it has been discovered in the mitochondria, where it participates in base excision repair (BER) [96]. Recently, the loss of the Fe-S cluster in Dna2 resulted in a change in conformation that caused it to impair DNA binding, which affected all its biochemical functions, such as its nuclease and helicase functions [97]. Additionally, iron transport and storage protein frataxin defects lead to BER defects. This suggests that the transportation and storage of iron from the labile iron pool are vital for DNA repair and replication enzymes such as Dna2 [98]. The helicases Dna2, FANCJ, ChlR1 and XPD have recently been discovered to have a Fe-S cluster and could be indirectly targeted by iron chelators that lower the available iron.

The four DNA polymerases: Pol $\alpha$, Pol $\delta$, Pol $\varepsilon$ and Pol $\zeta$, require iron as a cofactor [99]. When DNA replicates, it starts with a primer; the DNA primase initiates an RNA primer before DNA Pol $\alpha$ synthesises the leading strand; this additionally occurs at each Okazaki fragment. After roughly 20 nucleotides of DNA have been added by Pol $\alpha$, the strand is recognised, facilitating the binding of proliferating cell nuclear antigen and the subsequent recruitment of Pol $\delta$ and Pol $\varepsilon$ [100]. Pol $\delta$ completes every Okazaki fragment; thus, Pol $\delta$ primarily works on the lagging strand [101]. Pol $\varepsilon$ is principally used in the leading strand, whereas Pol $\zeta$ is used when DNA is damaged [100]. The polymerase Pol $\delta$, Pol $\varepsilon$ and Pol $\zeta$ have all been shown to have a Fe- S cluster in the 4Fe-4S arrangement that plays a crucial role in DNA replication [102–104]. Although its full role has not been fully elucidated, Fe-S coordination defects in these polymerases cause loss of function or deleterious DNA

replication. Whereas the Pol α Fe-S cluster has only been demonstrated in purified yeast and samples with primase, there is little structural information on the Fe cluster in Pol α. This could be due to the difficulty in purifying these enzymes without damaging the Fe-S cluster [102,105].

### 4.2. Ribonucleotide Reductases

Ribonucleotide reductase (RnR) is an essential enzyme in DNA replication and repair, reducing ribonucleotide 5′ diphosphates to deoxyribonucleoside 5′ diphosphates [106]. In cancer, there is an increased requirement for RnR highlighted by an increased DNA synthesis rate compared to non-neoplastic cells [107]. There are three classes of RnR: the predominant one in eukaryotes is class 1 Ia RNR, which contains a di-iron centre that maintains a diferric-tyrosyl radical ($Fe_2^{III}Y\cdot$) [108]. Class 1 Ia RnR has a large R1 and small R2 subunit; the small R2 subunit contains this di-iron centre required for catalysis [100]. RNR is able to redox cycle between the two iron states; this gives it its function by allowing RnR to function as an electron donor or acceptor [109]. An important part of retaining genomic integrity is ensuring there are normal levels of dNTPs, therefore deregulation of RnR is mutagenic. As RNR is directly dependent on the labile iron pool and intercellular iron, the dysregulation of iron levels can thus be mutagenic [110]. RnR also produces uridine diphosphate (dUDP), used in double-strand break repair. The disruption of RNR in tumours was shown to prevent thymidylate kinase from stopping dUDP from repairing DNA [111]. Therefore, an imbalance in RNR and dNTPs has resulted in DNA mutations and cell death [112,113]. This imbalance could lead to genomic instability in cancer cells and potentially increase survival by increasing DNA repair [5].

Iron Chelators and Their Activity against RNR

One hallmark of cancer is its ability to divide rapidly; for this to occur, neoplastic cells would need dNTPs in much greater volumes [4,5]. Thus, the deficiency of RNR function, which could result from an iron chelator, would result in cell cycle arrest in the $G_1$/S phase [114]. To that end, several iron chelators have been designed for cancer therapy to inhibit RnR: hydroxyurea, desferrioxamine, thiosemicarbazone derivates and pyridoxal isonicotinoyl hydrazone (PIH) have all been tested against cancer [115–118]. Hydroxyurea was the first metal chelator to be used to inhibit RNR. This is achieved by attacking the di-iron centre and reducing the tyrosyl radical via one-electron transfer, thus depleting the dNTPs pools and causing DNA damage [110,119]. This chelator is not iron specific, attacking other metallocentres, which produced varied results, with its anti-tumour capabilities being somewhat controversial [120]. This led to more iron-specific chelators, the most notable being desferrioxamine which is used clinically in iron overload diseases and is also researched for its anti-cancerous properties [121].

Desferrioxamine is a bacterial siderophore that binds iron in the labile iron pool, decreasing the available iron content to the cell [122]. High doses of Desferrioxamine can also disrupt iron homeostasis by removing iron, causing cell cycle arrest and enhancing apoptosis in cancer cells [123–125]. RnR, as described earlier, relies on the labile iron pool for the biosynthesis of its cofactors. Desferrioxamine complexes are very stable, leading to a full co-ordination sphere 1:1complex. As previously stated, they do not have an unbonded co-ordination site for Fenton chemistry [126]. Thus, desferrioxamine inhibits RnR's function because it removes the available iron from the labile iron pool. This reduction in labile iron stops the tyrosyl radical forming in the di-iron centre of RnR [121]. However, desferrioxamine has several drawbacks as an RnR inhibitor: poor membrane permeability, a high cost and a short life in the bloodstream [127]. Nonetheless, its limited success encouraged the design of new iron chelators for RnR inhibition, such as PIH and triapine (3-AP).

PIH is a tridentate chelator with hard oxygen donor atoms; as stated above, it prefers stronger cations as it's a strong donor and thus prefers $Fe^{III}$ over $Fe^{II}$ [128]. It is also much more readily able to penetrate the cell membrane at physiological pH as it is very

lipophilic [129]. PIH had three series created which depended on the number of ligands attached; they were known as the pyridoxal benzoyl hydrazone analogues (100 series), the salicylaldehyde benzoyl hydrazone analogues (200 series) and 2-hydroxy-1-naphthylaldehyde benzoyl hydrazone analogues (300 series) [130]. PIH and all analogues are tridentate ligands that bid in a 2:1 ligand to iron complex [131]. The 300 series was shown to be the most lipophilic and thus could penetrate the most readily and had the greatest efficacy at inhibiting RnR [127,132]. An example of this is the iron chelator 2-hydroxy-1-naphthaldehyde isonicotinoyl hydrazone; NIH, which has been shown to inhibit tumour growth in cell lines of breast, bladder and neck and also the CCRF-CEM cell line [133]. The main mechanism accredited for this was the inhibition of the tyrosyl radial in RnR, its effectiveness was shown in that it had an IC50 20-fold lower than desferoxamine. However, studies show redox cycling did not play a large role in NIH's antiproliferative activity [134]. This suggests that it inhibited RnR's function either directly or by removing iron from the labile iron pool.

Triapine (3-AP) is an iron chelator synthesised as a potent RnR inhibitor. It is used clinically for CML and is in clinical trials for aggressive myeloproliferative neoplasms, cervical cancer and solid-state tumours [135,136]. It has shown to be more potent than Hydroxyurea but appears to use a different mechanism as it works on Hydroxyurea resistant cells [137]. As triapine is redox-active it produces ROS, which is thought to make the di-iron centre of RnR unstable and inactive [25,138].

RnR is required for DNA replication and repair and has thus been a target for therapeutic agents since it was first discovered. Understanding its biosynthesis and metallocofactor assembly lead to the use of more iron chelators that target the iron core of RnR [40,139]. These iron chelators advanced over the years by taking advantage of how the design could further affect its function, inhibiting RnR. Iron chelators that could redox cycle have been used most recently in cancer cell lines as they can work in a variety of ways, including removing iron from the labile iron pool and interfering with the tyrosyl radical in RnR via the production of ROS [140,141]. This makes chelators an attractive option for future RnR inhibition and cancer therapy.

### 4.3. Iron Chelators and The Cell Cycle

Many iron chelators exert their antiproliferative effect by targeting key molecules in the cell cycle, such as RnR, cyclins, cyclin dependant kinases (CDKs), p21, hypoxia-inducible factor-1a (HIF-1a) and N-myc downstream regulatory gene-1 (Ndrg-1) [142–144]. The cell cycle relies on the binding of cyclin molecules and the corresponding cyclin-dependent kinase for the cell cycle to progress. Many iron chelators have been shown to affect cyclins and CDKs. One of the early studies exhibiting this used the iron chelator mimosine, which resulted in a marked decrease in cyclin D/CDK4 proteins in MDA-MB-453 human breast cancer cells [145]. This effect was then shown with the pyridoxal isonicotinoyl hydrazone class of iron chelators, particularly 2-hydroxy-1-naphthylaldehydeisonicotinoyl hydrazone (311), which showed a decrease in cyclins D1, D2, and D3 in SK-N-MC neuroepithelioma cell [146]. This has more recently been demonstrated with the iron chelator 311 and 4-[3,5-bis-(hydroxyphenyl)-1,2,4-triazol-1-yl]-benzoic acid (ICL670), which resulted in inhibition of the cell cycle by deregulating CDK2 and CDK9 [147]. Deferasirox has also demonstrated it can induce cell cycle arrest by downregulating cyclin D1, cyclin B, and CDK4 expression in the gastric cancer cell lines of AGS, MKN-28, SNU-484, and SNU-638 [148]. Deferasirox was also shown to upregulate Ndrg-1, p21, p27, and p53 expression. Iron chelators can cause cell cycle arrest; this exact mechanism is hard to pinpoint. It could be assumed that it is a general mix of iron deprivation in crucial enzymes mentioned above or specific iron chelators causing an increase in ROS, resulting in double-strand breaks in DNA and the resulting cell cycle arrest.

### 4.4. Could Iron Chelators Passively or Actively Target Fe-S Clusters

Iron is crucial for maintaining genomic stability in DNA polymerases and helicases through the Fe-S cluster and thus in DNA replication and repair. It therefore makes sense

that loss of chelate-able iron via iron chelators could limit the iron available for Fe-S clusters. Iron chelators have been shown to remove iron from the labile cytosolic iron pool and the mitochondrial labile iron pool, supplying iron to Fe-S clusters. As a result, it is logical that chelating iron would limit the iron available for DNA replication and repair. It is not fully understood to what degree iron chelators could remove iron from Fe-S clusters. Chelators usually work on the cheatable iron pool; however, the IRP sensor is an example of the removal of iron from Fe-S clusters. In iron-deficient conditions, IRP1 does not have enough iron to form a full [4Fe-4S], creating a [3Fe-4S] cluster instead. This degrades to the apo-protein of IRP1, which binds to IRE to produce more iron [52,149]. Further, the recent discovery that scaffold proteins such as ISU enable labile Fe -S clusters to form before incorporation into proteins raises many questions. Can iron chelators interfere with this pathway and potentially further limit Fe-S clusters' role in the DNA damage response? Currently, it is unknown and unlikely that iron chelators can directly interact with Fe-S clusters. However, it has been shown that Fe-S clusters incorporated into proteins, such as ribonucleotide reductase, can be inhibited by iron chelators that remove iron from the labile iron pool and lower the supply of iron into these proteins [110].

Fe-S clusters have inherent susceptibility to oxidation, which makes Fe-S targets for activation by ROS. This further mechanism iron chelators can be designed to target Fe-S clusters by designing chelators that can increase ROS. Fe-S susceptibility to ROS was first seen in antibiotics, where quinolones increase the NAD+/NADH ratio resulting in ROS production and Fe-S cluster disruption [150–152]. This is thought to be because the superoxide anion ($O_2^{\bullet-}$) and hydrogen peroxide can inactivate Fe-S clusters [153]. Since then, drugs that induced ROS, such as β-Phenethyl isothiocyanate (PEITC), were used to target Fe-S centres, specifically the NADH dehydrogenase Fe-S protein-3, which resulted in its degradation by lowering the mitochondrial respiration and mitochondrial glutathione [154]. However, the drug hydroxyurea is known to chelate iron, remove it from the labile iron pool, and inhibit RnR; it has been shown to target Fe-S clusters by inducing ROS [155,156]. It was postulated that the hydroxyurea effect could also be because it scavenges tyrosyl radicals in Fe-S biosynthesis [156].

Recently DFO (deferoxamine) has been altered so that it targets the mitochondria by the addition of a triphenylphosphonium group [157]. This resulted in the new complex, mitoDFO, accumulating in the mitochondria and lowering Fe-S cluster biogenesis. The diminished Fe-S cluster biogenesis consequently resulted in a decrease in the activity of the resulting Fe-S proteins. MitoDFO showed a 40-fold increase in antiproliferation of cancer cells. This is one of the first studies to examine how iron chelators targeted to the mitochondria can affect Fe-S clusters.

This is an evolving field; iron's crucial role in genomic stability is well-defined when referring to RnR; however, little is known about the effect that chelators or iron deficiency has on Fe-S clusters and nuclei acid metabolism [158]. It is established that the overexpression of key Fe-S clusters promotes tumorigenesis, and mutations in the DNA repair enzymes can cause cancer [79]. The susceptibility of these proteins to ROS raises whether iron chelators that are known to induce ROS and limit the labile iron pool could potentially target Fe-S centres in cancer. As iron chelators have been used in cancer for an extended period, it is unlikely that they alone would be the sole solution to the complex problem of iron metabolism, its altered role in cancer and targeting Fe-S centres by generating ROS. Although the use of iron chelators in combination with drugs designed explicitly for targeting Fe-S centres has not been studied; it could be noticeably synergistic as a cancer therapeutic. Iron chelators that are designed in the future to target Fe-S centres could benefit from targeting the mitochondria which requires iron for protein biogenesis and is susceptible to ROS.

## 5. Iron, ROS and Iron Chelators That Can Cause DNA Damage in Cancer

ROS in normal cells can be beneficial for signalling in cell defence or development; ROS mainly arises in the mitochondria as a by-product of oxygen metabolism by oxygens

incomplete reduction [158]. Low levels of ROS in normal cells contribute to the cell's proliferation, whereas ROS can contribute to apoptosis at intermediate levels. High levels of ROS contribute to necrosis; however, high levels can also be kept in check by antioxidant enzymes such as glutathione peroxidase [159]. In cancer, sustained growth signalling and proliferation factors are constantly turned on; consequently, there is an increased metabolic rate and hypoxia, resulting in heightened levels of ROS in cancer cells [160]. Cancer cells can also adapt to higher concertation of ROS by producing more antioxidants. The more elevated ROS occurring during oncogenic signalling by c-Myc are typically just under levels that would cause toxic levels of damage [161]. Cancer is known to rapidly grow in response to iron levels which enable DNA replication. Cancer cells therefore have an increased labile iron pool, and this resulting iron participates in the Fenton reaction to produce toxic radicals, contributing to a higher basal level of ROS. These high levels of iron can result in genomic instability due to the production of hydroxyl radicals that can damage DNA [162,163].

Hydroxyl radicals produced by the Fenton reaction can react with membrane phospholipids causing lipid peroxidation. The hydroxyl radical can react with the fatty acid chain, generating a hydroperoxidised lipid and an alkyl radical, which can cause further damage [10]. Lipid peroxidation can result in cytotoxic aldehydes such as malonaldehyde and hydroxynonenal, which can react with the amino group in DNA bases to form adducts [164]. The hydroxyl radical from the Fenton reaction leads to hydrogen atom transfer (HAT) in deoxyribose, occurring at different carbon sites on the deoxyribose. ROS can react with the nitrogen bases in DNA, resulting in DNA fragmentation and causing issues involving the coiling of DNA in chromatin [165]. On pyrimidine, this often occurs at the 5, 6 pyrimidine bond and results in the HAT from the methyl group. In thymine, the hydroxyl radical can also react with the methyl group leading to 5-(uracily)methyl radical [166]. This can react with other bases such as adenine and guanine to produce interstrand cross-links. The most studied and common example of DNA damage by ROS is the addition of the hydroxyl radical with carbon 8 on guanine generating 8-hydroxy-7,8-dihydroguan-8-yl radicals. This either undergoes oxidation to form 8-oxo-guanine or reduction resulting in the imidazole ring opening and formation of N5-substituted formamidopyrimidine (N5-R-FAPy) lesions [167]. DNA damage can produce replication errors, aberrant transcription or signalling pathways, and, in theory, results in greater mutations and clonal evolution. This is also aided by the fact that high volumes of ROS can contribute to mitogenic signalling giving tumour cells a further growth advantage. An example is that the oncogenic KRAS requires ROS for proliferation [168].

ROS can cause changes to DNA and thus the genome leading to changes downstream in metabolic pathways or cell signalling, potentially favouring sustained proliferation. Mitochondrial DNA is even more susceptible to ROS as it has no histones and cannot participate in nuclear excisions repair [159]. This, in part, has given rise to the 'horse and cart' model. The model links mitochondrial metabolism (the 'horse') to (the 'cart') gene expression, where interconnected reactions in the mitochondria can lead to changes in gene expression [169]. When the balance shifts to favour ROS, it creates an environment that lends itself to genetic instability and self-deterioration. The mitochondrial labile iron pool is increased in cancer to fill the need for the incorporation of iron into crucial clusters. The increased ROS and increased labile iron pool would theoretically result in more Fenton chemistry. This could in turn additionally increase the labile iron pool further as $O_2^{\bullet -}$ and $H_2O_2$ can oxidise aconitase facilitating its change from aconitase to IRP-1 with the release of $Fe^{2+}$ from the cluster [46]. Further, IRP-1 would bind to IRE resulting in more influx of iron.

## 5.1. Iron Chelators and ROS

There can be confusion regarding iron chelators' exact role and purpose when considering iron chelators and ROS. If iron chelators remove iron, they must reduce ROS and return to homeostasis. This is true for iron chelators that do not result in redox-active complexes. Hexadentate chelators are often used to treat iron overload, such as Deferoxamine, as they

are not redox-active. Iron chelators that form redox-active complexes, often bidentate and tridentate iron chelators, have free sites for redox cycling and thus can promote free radical generation [128]. Therefore, the stichometry of iron to the chelator is essential. This was shown in the iron chelator 2-dimethyl-3-hydroxypyrid-4-one, which potentiated $H_2O_2$ oxidative DNA damage; however, when prolonged exposure was maintained, it resulted in a protective effect in normal liver cells [170]. The iron complex formed after iron chelation is vital; in some cases, it can act as a reductant or an oxidant. This can be due to the iron complex's localisation in different cellular organelles. This, in theory, presents two ways to approach ROS in cancer, even more specifically when addressing iron chelation for cancer. The first method is to either reduce ROS and inhibit ROS signalling by chelating iron and preventing Fenton chemistry. The second method increases ROS by using an iron chelator with a redox-active complex to selectively kill cancerous cells [160]. The latter approach can be seen in Figure 5, where the iron chelator increases ROS via redox-active iron complexes resulting in DNA damage. All iron chelators discussed that induce ROS have their chemical structure and iron binding groups displayed in Supplementary Figure S1.

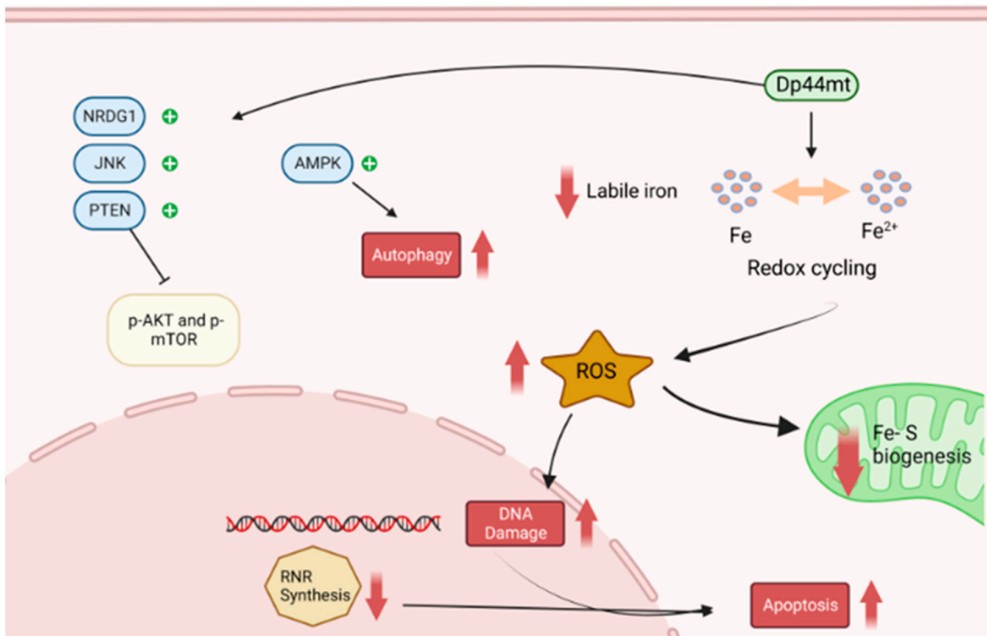

**Figure 5.** The potential therapeutic effect of redox-active iron chelators in cancer increasing ROS resulting in DNA damage while reducing the size of the labile iron pool in theory and thus iron available for Fe-S biogenesis and RnR (ribonucleotide reductase) synthesis. The iron chelator Dp44mt for example also upregulates NRDG1 and other cell signalling pathways.

5.1.1. Mimosine and Analogues

Mimosine (β-N-3-hydroxy-4-pyridine-a-amino-propionic acid) is a natural iron chelator and was initially used to synchronise cells in the G1/S phase of the cell cycle. It has been shown to complex iron and indirectly inhibit RnR, first demonstrated in viral DNA [171]. Due to its inhibition of RnR and its antiproliferative effect, the molecular mechanisms were studied on U937 leukaemia cells and promyelocytic HL-60 cell lines. Here, mimosine induced ROS, resulting in apoptosis [172]. This was confirmed using the powerful antioxidant N-acetyl cysteine (NAC), which inhibited ROS-induced apoptosis [173]. When NAC, a glutathione precursor, was used, no superoxide was produced. Hallak et al. theorised that as mimosine reduced glutathione, NAC reversed the apoptotic effect of Mimosine [173]. Furthermore, mimosine's ability to induce ROS was due to the inhibition of catalase [173]. Mimosine had previously been shown to induce DNA breaks; the study by Hallak et al. theorised that ROS induction causes this [173,174]. In another study, mimosine was tested against the cell proliferation of the C6 glioma cell line. The study found 12.6% apoptosis in

these cells compared to the control after 24 h at 250 μM [175]. Their findings exhibited an increase in ROS generation in the mitochondria and led to downstream upregulation of Jun-N-terminal protein kinase (JNK) and p38, which both play roles in apoptosis induced by stress [176]. This provided the first evidence that ROS generated after mimosine treatment was produced in the mitochondria, which differs from Hallak's findings [173]. More recently, mimosine has been shown to inactivate aconitase when complexed to ferrous iron. This ensues in the autooxidation of $Fe^{2+}$ and result in the formation of 8-hydroxy-2′-deoxyguanosine in Calf thymus DNA [177]. However, reports of mimosine's cytotoxic side effects have resulted in its limited use and the generation of prodrugs based on mimosine's structure [178]. Mainly a drug termed L-SK4, a methylated analogue of L-mimosine, compound 22. L-SK4 produced a four-fold induction of ROS after 24 h compared to the untreated control (100 μM) in A375 cells. They then go on to show that both the intrinsic and extrinsic pathways of apoptosis were activated suggesting that this was in response to ROS induction by L-SK4. Later studies showed that the ROS induction by L-SK4 resulted in a significant increase in sphingolipids which can act as messengers for both apoptotic pathways [179].

### 5.1.2. Di-2-Pyridylketone Isonicotinoyl Hydrazone (PKIH)

Di-2-Pyridylketone Isonicotinoyl Hydrazone (PKIH) analogues were designed after the 2-Pyridylcarboxaldehyde Isonicotinoyl Hydrazone Series. They are tridentate chelators that can form octahedral complexes with Fe (II) and use the nitrogen in the imine group and the carbonyl oxygen as donor atoms [180]. PKIH series were much more selective iron chelators than their predecessors. Compared to other previous iron chelators PKBBH, PKBH and PKTH, it had the greatest effect inhibiting the proliferation of SK-N-MC neuroepithelioma cells with IC50 values of 1–3 μm while having the weakest effect on the non-cancerous fibroblasts [181]. PKIH was shown not to increase ascorbate oxidation; it did result in DNA degradation in the presence of hydrogen peroxide by stimulating benzoate hydroxylation and increased intracellular ROS [182,183]. Interestingly, these studies showed that its cytotoxic effect could be reversed by adding catalase, a potent antioxidant.

### 5.1.3. DP44mT and Analogues

One of the most widely researched iron chelators that produces toxic levels of ROS is the tridentate iron chelator di-2-pyridyl ketone-4,4,-dimethyl-3-thiosemicarbazone (Dp44mT). Dp44mT binds to iron in the 2:1 ratio and is more selective for Fe(III). It is a member of the thiosemicarbazones chelator family in which triapine and pyridoxal isonicotinoyl hydrazone (PIH) had relative success; Dp44mT is an analogy of these two thiosemicarbazones [127,132,184]. Dp44mt contains the donor atoms of imine, nitrogen, sulphur and pyridyl [33]. It was initially shown to induce apoptosis in lung carcinoma in vivo (up to 47% against the control). They suggested that its cytotoxicity could result from its redox activity [185]. Additionally, they showed that the incubation with Dp44mT for two hours increased ROS. Later, its antiproliferative effect was seen by inducing DNA double-strand breaks culminating in cell cycle arrest [184]. It also exhibited selective inhibition of top2A, demonstrated in breast cancer cells and overcoming the altered autophagy in cancer cells [184,186]. It has since been used in a leukemic cell line, melanoma mouse xenograft, and osteosarcomas, where it increased levels of ROS [187–191]. Dp44mt also significantly increased autophagy via activation of the AMPK pathway whilst inhibiting metastasis of the Wnt/β-catenin signalling pathway in several cancer cell lines [191]. Dp44mt has been shown to increase the tumour suppressor phosphate and tensin homolog (PTEN) and NRDG1 [190]. This was shown to reduce p-AKT levels and has been repeated in pancreatic cancer [190,192]. Dp44mt has also been shown to inhibit mTORC1 in a ROS independent fashion, reducing p-AKT in A549 and A427 lung cancer cell lines [193]. The ability of Dp44mT to induce ROS has been shown to be crucial to its cytotoxic profile [194]. Dp44mT has high antiproliferative and cytotoxicity; however, it can induce cardiotoxicity if used at non-optimal doses. This cardiotoxicity required the design of second-generation

iron chelators, the main compound being di-2-pyridylketone 4-cyclohexyl-4-methyl-3-thiosemicarbazone (DpC) [195].

DpC, unlike Dp44mT, can be given intravenously and orally. It has demonstrated many of the same effects that Dp44mT has, such as overcoming chemotherapy drug resistance and increasing the expression of NDRG-1 without the oxidation of oxyhemoglobin and cardiac toxicity [196,197]. DpC, like Dp44mt, is tridentate and has soft donors such as N and S in the coordination sphere. This allows complexes to participate in redox reactions and has exhibited an increase in ROS; this aids its antiproliferative effects. It was recently shown to inhibit neuroblastoma growth in vitro and in vivo [198]. DpC has also recently been shown to inhibit TGF-β and Wnt/β-catenin signalling in pancreatic cancer, thus inhibiting the cross talk between pancreatic cancer cells and pancreatic stellate cells [199].

### 5.1.4. Di-2-Pyrineketone Hydrazone Dithiocarbamate (DpdtC)

The iron chelator Di-2-pyridine ketone hydrazone dithiocarbamate (DpdtC) was designed originally as an analogue of Dithiocarbamates. It was explicitly intended for iron chelation and thus had a strong chelating ability for metals. It was suggested that it chelates labile iron forming a redox-active complex that undergoes Fenton chemistry. DpdtC induced cell cycle arrest and DNA fragmentation, resulting in autophagy and ROS-mediated apoptosis in HepG2 cells [200]. A second study was then performed, studying the mechanisms of how this iron chelator exerted its anti-cancerous effect through the induction of ROS. They discovered that it stemmed from catalase inhibition by causing a conformational change in catalase [201]. This design shows that when considering the design of iron chelators, attention needs to be paid to their steric effects as they can also have an allosteric or physical effect. Later, this iron chelator exhibited that it could upregulate NRDG-1 by depleting the intracellular iron content, which was shown explicitly in HER-2 overexpressed cancer [202]. Later, it was demonstrated that ferritophagy, the process of ferritin degradation in the lysosomes, leads to ROS generation and lipid peroxidase [176]. DpdtC also inhibits the cell signalling pathway EGFR/AKT and induces apoptosis in KYSE-150 and KYSE-450 esophageal cancer cells [203].

### 5.1.5. KS10076

KS10076 a metal chelator that readily binds with iron, zinc and manganese has recently been shown to be redox active and therefore increase ROS production in cancer cells [204]. KS10076 was shown to decrease labile iron and ferritin resulting in an increase in IRP 2. KS10076 addition in SW480 cells CAKI1, COLO205, and HCT116 cells resulted in an increase in mitochondrial ROS and cellular ROS, whereas cell lines that were less sensitive to KS10076 had no increase in ROS, verifying that its cytotoxic potential is in the ability of it to form redox active complexes that can damage the cell. KS10076 induction of ROS results in the removal of aldehyde dehydrogenase isoform 1-positive (ALDH1$^+$) cancer stem cells.

### 5.1.6. Triapine and Deferiprone

Triapine is a tridentate iron chelator initially designed to inhibit RnR, discussed above, but it is also involved in Fenton chemistry. It has demonstrated it can produce hydroxyl radicals as its Fe complexes can participate in redox cycling, and thus it was shown to result in breaks in plasmid DNA [205]. It is mainly used in combination therapy as treatment by itself has not been as successful in clinical trials [206]. Deferiprone is one of the most commonly used orally active tridentate iron chelators; it has been used against prostate and cervical cancer cell lines. It is largely redox-inactive due to its low redox potential resulting in the inhibition of the Fenton reaction and lipid peroxidation [207]. Despite this deferiprone can also under certain conditions be pro-oxidant reacting where iron transfers its electron to deferiprone starting a cascade of redox reactions [177,207]. In cancer, deferiprone originally was shown to inhibit growth and induce apoptosis [1]. Subsequently, deferiprone demonstrated it could cause one-electron reduction to form superoxide, which can inactivate aconitase and play a role in Fenton chemistry [177]. Deferiprone has also

been shown to target cancer stem cells in MCF7 and T47D breast cancer cells, targeting mitochondrial oxygen consumption and glycolysis and increasing ROS production [208]. These iron chelators like the ones above have this ability to redox cycle and produce ROS under certain conditions. These chelators have found great use in cancer, as they can tip the scale to a toxic amount of ROS while limiting iron available.

## 6. Perspectives and Conclusions

It is clear that altered iron metabolism occurs in cancer and leads to increased labile iron pools in the cytosol, mitochondria and lysosomes. The mitochondrial labile iron pool provides iron to essential Fe clusters involved in DNA replication and repair, which aids in sustained proliferation. Nonetheless, Fe-S clusters are more susceptible to ROS, which can impair the function of proteins with Fe-S clusters. Iron chelators have demonstrated that they can remove iron from the labile iron pool in the mitochondria and cytosol. This results in less iron being available for incorporation into Fe-S clusters for DNA replication and repair. The most obvious example is RnR inhibition, which is one mechanism by which chelators can inhibit tumour cells. More recently, iron chelators have been designed for various purposes. These purposes include: inhibiting topoisomerase, targeting specific DNA repair enzymes, and causing DNA breaks and apoptosis, to name a few [69,161]. Iron chelators have also been designed to cause DNA damage by inducing ROS, which causes cell cycle arrest that inhibits cancer [176]. This expands the areas which need to be researched, such as, can iron chelators that produce ROS target Fe-S centres by inducing ROS whilst also limiting the labile iron pool? Thus, it would target the labile iron pool, i.e., the iron available for incorporation into these Fe-S clusters and the clusters themselves. Whether iron chelators can affect Fe-S clusters in necessary repair and replication enzymes, it is already clear that iron chelators are an attractive therapeutic option by limiting iron and reducing or increasing ROS. Iron chelators tackle altered iron metabolism in cancer by many different mechanisms in cancer cells, such as targeting proliferation, migration and invasion.

The field of iron chelators started by addressing iron overload; thus, iron chelators were not explicitly designed for cancer. A crossover was noticed in that most iron chelators could inhibit RnR and therefore had therapeutic potential in cancer. More recently, iron chelators are starting to have more significant design considerations for those that can induce ROS either by inhibiting antioxidants or resulting in redox-active complexes culminating in DNA damage. Iron's role in cancer is deeply connected to many steps, with enhanced iron being a characteristic of most cancer cells due to the high demand for DNA replication. The increased amount of labile iron in cancer cells can result in ROS and genomic instability, ensuring an environment where tumour cells thrive via mutations and evolution. That being said, knowing iron's altered and crucial role in cancer, its essential role in DNA replication and repair through key enzymes and its role in causing genomic instability, there needs to be greater emphasis on creating iron chelators that can induce ROS for cancer, or that can target the mitochondria; specifically, the site of Fe-S biogenesis. The use of redox active iron chelators that could specifically target the mitochondria could increase the potency of these drugs for targeting cancer. Further research still needs to be completed in using iron chelators in combination therapy with synergistic compounds that can aid in ROS development.

**Supplementary Materials:** The following supporting information can be downloaded at: https://www.mdpi.com/article/10.3390/app121910161/s1, Figure S1 shows redox active iron chelators mentioned in this paper their structure and iron binding groups.

**Author Contributions:** A.C. wrote this manuscript. S.V. and S.R. reviewed and edited this manuscript. All authors have read and agreed to the published version of the manuscript.

**Funding:** This research received no external funding.

**Institutional Review Board Statement:** Not applicable.

**Informed Consent Statement:** Not applicable.

**Data Availability Statement:** Not applicable.

**Acknowledgments:** We would like to thank Biorender for the provision of their website for construction of our figures.

**Conflicts of Interest:** The authors declare no conflict of interest.

## Abbreviations

| | |
|---|---|
| reactive oxygen species | ROS |
| ribonucleotide reductase | RnR |
| transferrin receptor 1 | TfR1 |
| divalent metal transporter1 | DMT1 |
| ferritin | Ft |
| ferroportin | FPN1 |
| iron regulating protein one and iron regulating protein two | IRP1 and IRP2 |
| iron-responsive element | IRE |
| outer mitochondrial membrane | OMM |
| inner mitochondrial membrane | IMM |
| Glutaredoxin-related protein 5 | GLRX5 |
| Iron sulphur cluster | ISC |
| DNA damage response | DDR |
| Nutrient-deprivation autophagy factor-1 | NAF-1 |
| transcription factor II H | TFIIH |
| Fanconi Anemia | FA |
| base excision repair | BER |
| uridine diphosphate | dUDP |
| pyridoxal isonicotinoyl hydrazone | PIH |
| triapine | 3-AP |
| cyclin dependant kinases | CDKs |
| hypoxia-inducible factor-1a | HIF-1a |
| N-myc downstream regulatory gene-1 | Ndrg-1 |
| 2-hydroxy-1-naphthylaldehydeisonicotinoyl hydrazone | 311 |
| 4-[3,5-bis-(hydroxyphenyl)-1,2,4-triazol-1-yl]-benzoic acid | ICL670 |
| deferoxamine | DFO |
| hydrogen atom transfer | HAT |
| β-N-3-hydroxy-4-pyridine-a-amino-propionic acid | Mimosine |
| N-acetyl cysteine | NAC |
| Jun-N-terminal protein kinase | JNK |
| Di-2-Pyridylketone Isonicotinoyl Hydrazone | PKIH |
| di-2-pyridyl ketone-4,4,-dimethyl-3-thiosemicarbazone | Dp44mT |
| di-2-pyridylketone 4-cyclohexyl-4-methyl-3-thiosemicarbazone | DpC |
| Di-2-pyridineketone hydrazone dithiocarbamate | DpdtC |
| aldehyde dehydrogenase isoform 1-positive | ALDH1$^+$ |

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
