# Peer review of "The Role of Iron in DNA and Genomic Instability in Cancer, a Target for Iron Chelators That Can Induce ROS"

_applsci, doi:10.3390/app121910161_

Round 1

Reviewer 1 Report

In this manuscript, the authors have discussed the role of iron in DNA (nuclear DNA and mitochondrial DNA) and genomic instability. Firstly, they described the properties of iron, including redox activity, sensitivity, thermodynamics, and lipophilicity. Next, they overviewed the function of iron metabolism in normal and cancer cells. Then, they discussed iron chelators, Fe-S clusters, and ROS, and their impact on DNA and genomic instability in cancer. Overall, they have fully described the function of iron metabolism and suggested iron chelators could be utilized to aid in combination therapy with ROS-induced compounds. 

Reviewer 2 Report

This review article is basically well written about overview of iron metabolism and role in DNA and genomic instability in cancer. Authors provided accurate information of iron metabolism and iron chelators based on Fe-S clusters. Iron chelators are known to inhibit the proliferation of cancer cells via apoptosis. Although ROS induction by iron chelators is not well known, it is very important points for many researchers. Moreover, iron metabolism of mitochondria has been paid attention recently. It’s worth to provide the information about mitochondria ROS and mitochondria targeting iron chelators. However, there are some concerns as follows.

Major comment

1. Authors mainly focused on Fe-S cluster, ROS, DNA damage and their related signals such as JNK pathway. However, iron chelators are known to affect other signal transductions in cancer cell. For example, PI3K-AKT pathway is effectively suppressed by some iron chelators though the pathway is often activated in many kinds of cancer. It’s helpful information when readers consider use iron chelators for cancer treatment.

2. Stemness is also important factor of cancer cell and thought to be related with iron metabolism and genomic stability. Iron chelator has unique function to inhibit stemness in cancer. Recent review article introduces the function (Szymonik J., et al. Int J Mol Sci. 2021;23(1):89). Please consider to reveal the relationship between iron metabolism, ROS and stemness maintenance in cancer.

3. Authors introduced some iron chelators such as DFO, DP44mt and DpdtC. However, the drugs were not included in the figure. I recommend to inert the drugs in the figure with work points, which is helpful information for readers.

Minor comment

1. Figure 3 is duplicated. Authors must modify it.

Reviewer 3 Report

In this review manuscript entitled “Iron’s role in DNA and genomic instability in cancer, A target for iron chelators that can induce ROS,” the authors attempt to understand the role of iron in cancer development and the therapeutic potential of iron chelators in cancer therapy. This manuscript is well written and should be accepted for publication in the journal in the better interest of readers of Applied Sciences.

Reviewer 4 Report

Excellent review which should assist the scientific community in combating cancer.

Reviewer 5 Report

In the proposed manuscript, titled “Iron’s Role in DNA and Genomic Instability in Cancer, A Target for Iron Chelators that can Induce ROS”, the authors describe the role of iron in the human body and focused on the role of iron in DNA damage and genomic instability.

The manuscript might be quite interesting and relevant to nanomedicine, nano-biotech, and, more in general, the medical community, however, it requires numerous modifications before publication in Applied science. There are some relevant formal and technical issues that must be addressed.

1.      From a formal point of view, the article must be fully revised. In my opinion, the introduction is poorly written: although correct information is reported, it is very poorly presented.

2.      The authors mixed two different reference styles. For instance, sometimes they report the references in round brackets, sometimes as a number. The authors must uniform the references style after careful consultation with the guide for authors.

3.      The quality of the few images in the manuscript is very bad. For example, in fig. 4, the letter “A” covers part of the picture. Moreover, I believe that the Figure reported on page 15 is Figure 5 instead of FIGURE 3.  

4.      The manuscript is rich in abbreviations and acronyms, some of which are commonly used in numerous scientific articles (ROS, DNA, OMM…) while others are more specific and less common. Usually, the use of acronyms should facilitate the reader but, not in this case. For me, it is mandatory to insert a list of acronyms at the beginning of the manuscript.

5.      In my opinion, the topic of the manuscript does not fit with the topic of the Journal. The authors should submit the manuscript to another journal.

Round 2

Reviewer 2 Report

This review can provide scientific information for readers. The revised process has been adequately performed.

Reviewer 5 Report

I greatly appreciated the changes made to the manuscript which improved the quality of the paper.